# The Role of Antibodies in the Treatment of SARS-CoV-2 Virus Infection, and Evaluating Their Contribution to Antibody-Dependent Enhancement of Infection

**DOI:** 10.3390/ijms23116078

**Published:** 2022-05-28

**Authors:** Mohammed A. H. Farouq, Reinaldo Acevedo, Valerie A. Ferro, Paul A. Mulheran, Mohammed M. Al Qaraghuli

**Affiliations:** 1Department of Chemical and Process Engineering, University of Strathclyde, 75 Montrose Street, Glasgow G1 1XJ, UK; paul.mulheran@strath.ac.uk (P.A.M.); mohammed.al-qaraghuli@strath.ac.uk (M.M.A.Q.); 2The Jenner Institute, University of Oxford, Old Road Campus Research Building, Roosevelt Drive, Oxford OX3 7DQ, UK; racevedo7810@gmail.com; 3Strathclyde Institute of Pharmacy and Biomedical Sciences, University of Strathclyde, 161 Cathedral Street, Glasgow G4 0RE, UK; v.a.ferro@strath.ac.uk; 4EPSRC Future Manufacturing Research Hub for Continuous Manufacturing and Advanced Crystallisation (CMAC), University of Strathclyde, 99 George Street, Glasgow G1 1RD, UK

**Keywords:** coronavirus, SARS-CoV-2, antibody-dependent enhancement, COVID-19

## Abstract

Antibodies play a crucial role in the immune response, in fighting off pathogens as well as helping create strong immunological memory. Antibody-dependent enhancement (ADE) occurs when non-neutralising antibodies recognise and bind to a pathogen, but are unable to prevent infection, and is widely known and is reported as occurring in infection caused by several viruses. This narrative review explores the ADE phenomenon, its occurrence in viral infections and evaluates its role in infection by SARS-CoV-2 virus, which causes coronavirus disease 2019 (COVID-19). As of yet, there is no clear evidence of ADE in SARS-CoV-2, though this area is still subject to further study.

## 1. Overview of SARS-CoV-2 Virus

### 1.1. The SARS-CoV-2 Virus Structure and Function

The causative agent of the COVID-19 pandemic is a previously unidentified virus strain, denoted as SARS-CoV-2 [1]. COVID-19 is generally associated with different symptoms including: fever, persistent dry cough, shortness of breath, and loss of taste and/or smell, and in many cases this disease has been fatal [2]. Consequently, the development of novel therapies has been a global priority for researchers. SARS-CoV-2 is composed of different structural proteins: the spike (S), membrane (M), envelope (E), and nucleocapsid (N) proteins [3]. SARS-CoV-2 infects human cells through binding of the virus receptor binding domain (RBD), located at the tip of the S protein (Figure 1), to the angiotensin-converting enzyme 2 (ACE2) receptor on human cell surfaces to facilitate the entry process and infection. Therefore, most anti-SARS-CoV-2 therapies have focussed on targeting the S protein, with the aim of inhibiting its binding to the ACE2 receptor.

Upon entry, two open reading frames (ORFs) 1a and 1b translate to two polypeptides, known as ORF 1a and 1b. This further encodes two proteases; the main protease (M^pro^), that is also identified as chymotrypsin-like cysteine protease (3CL^pro^), and papain-like protease (PL^pro^) [4]. The polypeptides 1a and 1ab invade host-cellular ribosomes to facilitate their translation, where they are processed by M^pro^ and PL^pro^, encoding several non-structural proteins (nsPs). The nsPs help the structural proteins (S, M, N, and E) enter the endoplasmic reticulum/golgi apparatus, and are involved in viral assembly and packaging [5]. The viral genome binds to the N protein, resulting in the formation of a ribonucleoprotein complex that facilitates viral replication [6]. NsP12 is an RNA-dependent RNA polymerase (RdRp), which plays a critical role in the assembly of the entire RNA polymerase replicative machinery, and is a key enzyme mediating the synthesis of all viral RNA molecules [7], making it a potential therapeutic target. In addition, guanine N7-methyltransferase (N7-MTase), found at the C-terminal of SARS-CoV-2 nsP14, is crucial for exonuclease activity [8]. Inhibition of this target could interfere with enzyme catalysis and prevent capping of the 5′-ends of viral genomic RNA and sub-genomic RNA, that is crucial in SARS-CoV-2 evasion of the host immune response [9]. Failure of RNA capping leads to viral RNA degradation, and interference with the replication cycle [10].

Other potential therapeutic targets include M^pro^ and PL^pro^, since their inhibition can stop the production of nsPs, which are critical to viral transcription and replication. Grotessi et al. [11] studied the conformational arrangements of M^pro^, and found that the protein is composed of three parts: domain 1 (residues 8–101), domain 2 (residues 102–184), and domain 3 (residues 201–303). The protease catalytic site is formed by a Cys-His dyad found in a pocket between domains 1 and 2. A molecular dynamics (MD) simulation study has predicted the interaction as an induced fit model [11]. Therefore, blocking the functional unit that cleaves the polyprotein could represent a compelling target for the development of new therapeutics. One possible method for blocking these targets, especially the S protein, could be achieved by antibodies.

### 1.2. Immune Responses to Viruses: The SARS-CoV-2 Case

The innate immune response (IIR) is the first line of defence against viral infection, and it is rapidly induced, but is of low specificity. Some evidence suggests the importance of IIR in early life when adaptive functions are not completely developed [12]. The innate immune cells express pathogen-recognition receptors (PRRs), such as C-type lectin receptors, NOD-like receptors (NLRs), RIG-I-like receptors (RLRs), and Toll-like receptors (TLRs) [13]. These receptors sense pathogen-associated molecular patterns (PAMPs) that include different viral structures like nucleic acid. RNA from coronavirus and other respiratory viruses are recognised by cytosolic and endosomal RNA receptors like RIG-I or TLR 3 and 7 [14]. Overall, the innate immune system senses foreign viral material and triggers downstream signalling, which induces transcription factors in the nucleus (NF-KB) that stimulate the expression of cytokines and interferons (IFNs) to fight against an infection [15].

IFNs play a major role in antiviral activities and are divided into three families (type I, type II and type III) based on their homology and biological activities [16]. Type I IFNs (e.g IFNα, INFβ, etc) are one of the first cytokines produced during viral infections, and their functions include inhibition of cellular translational function to prevent virion release, prevention of virus entry, and inhibition of virus transcription [17]. Type II IFNs are structurally unrelated to the other two classes of IFN genes and are produced by NK cells during antiviral IIR. IFN-γ improves the antiviral effect of NK cells and macrophages, but also has a great impact on the maturation and activation of antigen presenting cells (APCs). This effect drives stimulation of the adaptive antiviral response to clear the infection and generate memory to further fuel responses against the pathogen [17].

Type III IFNs also have an important role in tissue-specific IFN responses, and recent studies have shown that type III IFNs are better than the other two types of IFN. Type III IFNs are lower in magnitude, less inflammatory, and concentrated in regions of anatomic barrier signalling [18]. The antiviral effects of type III IFNs are especially evident at epithelial barriers, such as the gastrointestinal, respiratory, and reproductive tracts [19]. Type I and III IFNs are also considered to be crucial in antiviral responses, and SARS-CoV-2 has been shown to be sensitive to pre-treatment with IFN-I and III in in vitro assays [20,21]. The suppression of IFN responses in the early stages of COVID-19, reduces the host’s capacity to eliminate the virus and its spread [22]. However, IFN responses may also cause immunopathies, mainly if the immune response is not activated at the right time or intensity [23]. Overall, the early stages of type I IFN deficiency and the late stage of IFN persistence, could be a hallmark of severe COVID-19 infection [19,20,21,22,23,24]. In contrast, type III IFN responses are restrictively mucosa-specific, and the antiviral defences are weaker than those induced by pro-inflammatory responses. So far, many studies have investigated the immunomodulatory and antiviral roles of both type I and type III IFNs in SARS-CoV-2 infection, and it is imperative for IFN-based prophylactic development [24].

The adaptive immune response (AIR) is considered the main mechanism to limit viral replication and spread. AIR is led by the induction of proinflammatory cytokines and the activation of CD4^+^/CD8^+^ T cells, and B cells [25,26]. Chen et al. (2010) demonstrated that CD4^+^ T cells were very important for controlling SARS-CoV replication in disease T cell knock out mice. Recently, T cell studies demonstrated that CD8^+^ and CD4^+^ memory T cells are induced in SARS-CoV 2 convalescent patients and may have a major role in host protection [27]. However, in COVID-19, total CD4^+^ and CD8^+^ T cells are significantly reduced [28]. The importance of T cell responses is notable in that T cells may be sufficient to clear the virus in the absence of antibodies [29].

The induction of neutralising antibodies is an important effector mechanism to clear infections against viral diseases like influenza, dengue and COVID-19. More than 90% of infected patients with SARS-CoV-2 seroconvert between 5–15 days and the primary antigen for seroconversion is the spike protein (S) from the virus envelope [30]. The receptor binding domain (RBD) of S protein is the target of >90% of neutralising antibodies in COVID-19 cases [31]. Interestingly, SARS-CoV-2 neutralising antibodies have little to no somatic hypermutation (a process in which point mutations accumulate in the antibody variable regions to enhance their ability to bind foreign pathogens) [32,33]. These data indicate that the development of neutralising antibodies against SARS-CoV-2 is triggered by T cell independent mechanisms, as it can be accomplished by many B cells with little or no affinity maturation required. Work by Anderson et al. (2020) [34] and Shrock et al. (2020) [35], suggests that neutralising antibody responses generally develop from naive B cells, and not from pre-existing memory B cells.

The T cell response against COVID-19 has been underestimated, and recent data shows its importance in inducing memory and highly protective response. SARS-CoV-2-specific CD4^+^ and CD8^+^ T cells have been correlated with reduced disease severity, while neutralising antibodies in the same individuals did not [36]. It is also known that administration of high doses of SARS-CoV-2 neutralising monoclonal antibodies into SARS-CoV-2-infected patients had relatively limited effects on COVID-19 in clinical trials [37,38]. Altogether, these data highlight the importance of inducing high quality antibodies and memory B cells for appropriate protection against SARS-CoV-2. Probably with the cooperation of CD4^+^ cells and induction of B cell responses through T cell dependent mechanisms. New vaccine candidates are focused on the use of immune stimulating adjuvants and vaccine formulations, to elicit long lasting and highly neutralising responses against the virus through T cell activation [39].

Coronaviruses infect cells by multiple approaches, mediated by direct/indirect antibody Fc receptors or through targeting of the RBD to the ACE2 receptor. Infection of immune cells like macrophages and mast cells is possible if they express the target receptor. In fact, the exacerbated activation of macrophages and mass induction of cytokines supports the pathogenesis and inflammatory reactions in the severe forms of the disease [40]. Results from Maemura et al. [41] suggest that the Antibody-dependent enhancement (ADE) mechanism requires both Fc receptor engagement and ACE2 activation, although it is not clear that ADE inducing antibodies have a direct role in stimulating the cytokine storm. On the other hand, the quality and quantity of the SARS-CoV viruses infecting macrophages seems to be more related to the hyper-inflammation process. Therefore, to discover the mechanisms underlying SARS-Cov2 cytokine production it is important to develop new drugs to block the pathological process [42]. In addition, many groups are focusing their efforts on developing new vaccines inducing T cell responses, which may be more effective in controlling and reducing the viruses in the host cells than the currently approved COVID vaccines, which mainly induce a neutralising antibody response [43].

### 1.3. The Role of Antibodies against SARS-CoV-2 Virus

Passive immunisation has been routinely used for over a century to prevent diseases such as rabies, hepatitis B, and tetanus [44]. Another aspect of protection is based on utilising antibodies to block virus infection by a process called neutralisation. Virus neutralisation inhibits the acquisition of a pathogen or limits its pathogenesis. This can be achieved through pre-attachment neutralisation, by antibody binding to micro-organisms, and aggregation of antibodies may allow more efficient entrapment of pathogens in mucous, enhancing their subsequent clearance [45]. Virus neutralisation can also take the shape of interference with the viral attachment through binding to ligands vital for attachment of the pathogen to its host receptor through steric hindrance [46]. Post-attachment neutralisation can be based on inhibition of fusion/entry [47]; or inhibition of other steps in the organism’s lifecycle once it has successfully entered host cells by engineering cells to express intracellular antibodies (intrabodies) [48]. Consequently, virus neutralisation by antibodies could be exploited to protect against the SARS-CoV-2 virus. NHS England, for example, has provided a list of symptomatic, non-hospitalised COVID-19 patients who are showing no sign of clinical recovery and have priority for treatment with mAbs [49].

Several biopharmaceutical companies and academic institutions have co-operated to develop monoclonal antibodies (mAbs) for the treatment of COVID-19 and protection from infection against the SARS-CoV-2 virus. These efforts were based on previous experience in developing mAbs against pathogens such as syncytial virus (Palivizumab) [50], Ebola virus (Ansuvimab-zykl) [51], and *Bacillus anthracis* (Obiltoxaximab) [52]. Numerous SARS-CoV-2 mAbs have entered clinical trials for the treatment of patients with varying degrees of SARS-CoV-2 infection. There are currently 39 antibodies at different stages of clinical trials (Table 1). On 9 November 2020, the U.S. Food and Drug Administration (FDA) issued an emergency use authorisation for Bamlanivimab for the treatment of mild-to-moderate COVID-19 in adult and paediatric patients [53]. Regeneron REGN-COV2 (Casirivimab and Imdevimab) was also approved by the FDA (on 21st November 2020) in adults with mild-to-moderate COVID-19 who are at high risk for poor outcomes [53]. However, in January 2022, the FDA has limited the use of Bamlanivimab and Etesevimab, due to the Omicron Variant, to only when the patient is likely to have been infected with or exposed to a variant that is susceptible to these two antibodies [1].

All these clinical candidates are full length IgG antibodies targeting the SARS-CoV-2 S protein. In addition to these antibodies, various antibody formats are currently in pre-clinical development, including human mAbs [54], human single domain antibodies [55], llama-derived nanobodies [56], shark single-domain Variable New Antigen Receptors (VNARs) [57], humanised nanobodies [58], and human Fab [59], bi-specific or tri-specific antibodies [60], and nucleic acid encoding antibodies [61]. In addition, the route of administration could play an important role to surpass SARS-CoV-2 infection. For instance, Halwe et al. [62] have examined DZIF-10c, a mAb that could be administered intranasally to provide both prophylactic and therapeutic effects against COVID-19. DZIF-10c was able to abolish the infectious particles in SARS-CoV-2 infected mice, and mitigated lung pathology when administered prophylactically. DZIF-10c has completed phase 1 and 2 trial in healthy volunteers and SARS-CoV-2 infected individuals (ClinicalTrials.gov Identifier: NCT04631705).

The current success in developing antibodies against SARS-CoV-2 in a relatively short period was supported through the significant development of several laboratory and computational techniques. The antibody development process relies on techniques such as single B cell sorting, phage display, development of transgenic mice, high throughput sequencing, and modelling and computational analysis [53]. These techniques have the privilege to generate antibodies with exceptionally high affinity and superior neutralising function. Most of the isolated neutralising mAbs are developed using single B cells from individuals infected with COVID-19, as comprehensively detailed in [2]. These mAbs are specific to the RBD of the SARS-CoV-2 S protein, to compete with the ACE2 receptor to bind the RBD and neutralise infection. Moreover, these antibodies can also be engineered to remove or change the sites related to potential side effects, or can be engineered into multivalent or multi-specific neutralising antibodies. The selected antibodies can also be manufactured on an industrial scale to meet increasing clinic demands. The shark VNARs and llama nanobodies represent promising approaches for various therapeutic applications, since they could bind to cryptic epitopes that are inaccessible to full antibodies [63]. These variable domains could additionally bypass any potential ADE associated with SARS-CoV-2 infection, as they lack the Fc domain. Therefore, they could be considered an effective therapeutic option against coronaviruses.

Nevertheless, there are issues that should be considered related to dosing instructions, potential side effects, and drug interactions. Possible side effects of Casirivimab and Imdevimab include: anaphylaxis and infusion-related reactions, fever, chills, hives, itching, and flushing [53]. In addition, viral diversity and mutations could lead to the emergence of viruses with resistance mutations under the selective pressure of mAb treatment [54]. The latter point could be surpassed by selecting antibodies that are being developed to target conserved regions of the viral S protein, and a combination of different mAbs targeting various sites on the S protein were also considered to overcome these anticipated mutations [13]. Certain types of anti-viral antibodies could also be involved in a phenomenon known as ADE, where antibodies could enhance the infection and increase the severity of the resulting disease [64]. ADE, however, is a very complex process that involves different sophisticated factors that require comprehensive understanding prior to concluding whether a specific antibody is involved in this process or not.

## 2. Antibody-Dependent Enhancement (ADE)

### 2.1. The Principle of ADE

Antibodies involved in ADE may be generated from a previous infection or vaccination, and although virus-specific antibodies generally play a crucial role in infection control, sometimes the antibodies may aid viral replication [65]. This could occur upon infection with a different strain of the virus, when antibodies from a prior infection are still circulating or are induced by stimulation of memory cells (mainly B cells). These non-neutralising antibodies may bind to the virus and improve their uptake into the host cells. Further intracellular signalling could promote replication of the virus (Figure 2), and, as the virions are taken up by Fc-receptor bearing host cells [66], this results in wide-spread amplification of the original infection, resulting in severe and life-threatening viral infection [65].

There are generally two mechanisms of ADE: (1) extrinsic ADE, which contributes to enhanced viral entry and occurs through Fc-receptor facilitated virus contact and entry following viral receptor-mediated endocytosis [68] and (2) intrinsic ADE, which results in increased virus production by inhibition of type1 interferon (IFN1) and activation of interleukin-10 (IL10) biosynthesis [69]. Various studies have acknowledged the involvement of Fcγ receptors (FcγRs), surface receptors expressed on immune cell surfaces to recognise the Fc portion of IgG antibodies and activate comprehensive effector functions, as crucial mediators of ADE [66].

### 2.2. ADE in Viruses

Numerous articles have examined the contribution of antibodies to ADE in various flaviviruses, such as Dengue virus, Zika virus, and the enhancement of Zika by West Nile virus (WNV) antibodies [70,71,72]. ADE was initially described in Dengue virus infection [70]. The mechanism of ADE in Dengue virus was proposed to be intrinsic, which includes alteration of innate immune effectors by internalised virus-immune complexes to favour enhanced replication and release [73]. The antibodies’ role was postulated when reports suggested that patients with a particularly severe Dengue virus infection were associated with prior infection with a different Dengue virus serotype and pre-existing, non-neutralising anti-Dengue virus antibodies [74]. Activating FcγRs, such as FcγRIIa and FcγRIIIa, could promote Dengue virus infection, while FcγRIIb could generate a negative regulator for the ADE process [75,76].

The envelope of the Dengue virus contains the surface proteins E and M. E glycoprotein has a crucial function in viral attachment to cells [77]. The M protein can be found in two forms: prM, an immature precursor of the M protein and mature extracellular virions with the M protein. PrM is cleaved during maturation to yield the full M protein, induced during primary Dengue infection; and prM specific antibodies are highly cross-reactive, infection-enhancing, and possess limited neutralisation properties [78]. In secondary infection, these antibodies can bind to the infecting virus and so are postulated to play a key role in Fc-mediated ADE [78]. FcγR-mediated entry and infection enhancement, as key immune evasion mechanisms for Dengue virus, were demonstrated both in vitro and in vivo [79,80]. In addition, abrogation of the Fc–FcγR binding resulted in diminishing the pathogenic activity of these antibodies [81], and patients with symptomatic Dengue virus infection possess elevated serum levels of Fc glycoforms, with improved affinity for FcγRIIIa [82], and increased allelic frequency affinity single nucleotide polymorphism (SNP) of FcγRIIa [83]. An SNP in FcγRIIa could result in altered affinity of the receptor for different subclasses of IgG antibodies, and is crucial in determining the protection from, or the susceptibility to, severe clinical infection of dengue. These results have collectively highlighted the role of activating FcγRs in modulating Dengue virus infection.

In addition to the role of antibodies in Dengue virus, Furuyama et al. [84] studied the mechanism of ADE in Ebola virus in vitro, and found that besides Fc-receptor mediated ADE, the virus could utilise complement component C1q for ADE of infection. The C1q mechanism is independent of Fc-receptor mediated ADE, and is driven by the cross-linking of virus-antibody C1q complexes to cell surface C1q receptors [84]. However, both animal and clinical studies have failed to support a pathogenic role for antibodies in these infections [66]. Administration of a subset of potently neutralising mAbs revealed no protection, and a subset of highly protective mAbs only showed moderate neutralising activity, suggesting neutralisation was not sufficient for protection or strong neutralising activity is not consistently essential for protection [85]. In addition, FcγR engagement was essential for their antiviral potency, as loss of their FcγR binding capacity was accompanied with considerable decline in their protective activity [86]. Furthermore, the therapeutic implementation of anti-Ebola virus mAb114, a single monoclonal antibody targeting the RBD of Ebola virus glycoprotein did not cause side effects related to disease enhancement [87].

Antibodies against Dengue virus have shown a wide variety of in vitro cross-reactivity with Zika virus, and West Nile virus (WNV), especially between the former [71,88,89,90]. Both viruses share immunodominant epitopes that can provoke cross-reactive T lymphocytes to create protective immune responses [91,92]. The enhancement of Zika virus infection by antibodies from humans with symptomatic or asymptomatic WNV infection was tested in vitro [72]. Antibody-positive sera failed to inhibit Zika virus, and was proposed to cause an enhancement of the infection, which was attributed to the presence of non-neutralising antibodies to E protein that bind the pathogen, but do not interfere with infectivity, stimulating Fc-receptor mediated ADE [72]. The majority of specific antibodies against Zika virus were able to increase heterotypic viral replication in vitro [93,94]. Moreover, this proposed ADE process was prevented through blockage of Fc-FcγR interaction by antibody mutation [95] or pre-treatment with α-FcR antibodies [95,96]. Likewise, antibodies against WNV have enhanced Zika virus infection [71]. These in vitro data, nevertheless, were difficult to be extrapolated in vivo, as animal studies have shown conflicting outcomes [71,95,97]. Consequently, these results exemplify the divergence between in vitro and in vivo evaluation that adds huge complexity to the ability to understand and confirm whether a specific antibody contributes to the ADE process.

### 2.3. ADE in SARS-CoV

ADE in coronaviruses could be driven by FcγR bearing cells, and involves the key structural proteins found in these viruses, especially the S and N proteins that represent the core mechanism through which viral entry and infection occur [98]. Therefore, it is important to study the interactions of these proteins when developing vaccines against coronaviruses to ensure maximum efficacy, and minimise the possibility of ADE. Antibodies generated by the human immune system in response to SARS-CoV infections target the S protein, and these may contribute to the Fcγ-mediated ADE of the SARS-CoV virus [98]. A study in mice has attributed ADE in SARS-CoV infection to be driven by IgG_1_ antibodies against the S protein, while anti-S protein IgG_2a_ antibodies neutralised the virus, without the generation of ADE [99]. A further study in mice showed that immunisation with the SARS-CoV S protein aggravated the infection by causing Th2-type immunopathologic lung symptoms [100]. In addition, Wang et al. [101] studied different peptides of the SARS-CoV S protein and concluded that peptide-based vaccines could be designed to avoid ADE via elimination of the S_597–603_ epitope. It is noteworthy that cytokines secreted by Th1 and Th2 cells mediate isotype switching to IgG_2a_ and IgG_1_, respectively [102]. In addition, the IgG_1_ and IgG_2a_ ratio denotes the balance between humoral immunity (Th2 based) and cellular immunity (Th1 based) [103].

The in vitro mechanism of ADE against SARS-CoV varies considerably from the well-established mechanisms that govern ADE in Dengue virus. The antibodies produced against the S protein could increase the binding of the virus to FcyRII receptors, and lead to enhancement of uptake by host cells [104]. Dengue virus ADE depends on activating FcγRs like FcγRIIa and FcγRIIIa, whereas ADE mediated by SARS-CoV mAbs is dependent primarily on the inhibitory FcγRIIb, and has been shown to cause preferential infection of B cell (cell lines) in vitro [75,105]. Viral subunit vaccines frequently comprise immunodominant epitopes that are non-neutralising and could divert host immune responses, and these epitopes should be excluded in any prospective vaccine design. For instance, a vaccine developed for Middle East respiratory syndrome coronavirus (MERS-CoV) used an immunofocusing approach, through testing of the neutralising immunogenicity index, to specifically target the S1 domain of the virus RBD protein [106]. This resulted in a better immune response than targeting the full S protein, with notably high titres of neutralising antibodies for the S1 domain of the S protein [106]. Wan et al. [107] also investigated the ADE mechanism for MERS-CoV, and the antibody/Fc receptor complex was found to mimic the viral receptor involved in mediating viral entry.

A novel mechanism for ADE occurrence was found to involve a neutralising antibody binding to the S protein [107]. This was thought to trigger a conformational change of the S protein and mediate viral entry into FcγRs expressing cells through canonical viral-receptor-dependent-pathways [107]. Despite this possible entry mode, SARS-CoV can bind with high affinity to its entry receptor ACE2, so it is uncertain whether the virus needs to utilise low-affinity FcγRs, like FcγRIIb, for infection within the lung microenvironment. In Dengue virus, the situation was different because the lack of a specialised high-affinity entry receptor could force the virus to utilise the FcγR pathway [66]. In contrast to Dengue virus infections, cells expressing FcγR cannot withstand productive SARS-CoV infection, as these cell types are not tolerant for viral replication [108]. Yip et al. [109] demonstrated the occurrence of ADE in SARS-CoV Spike-pseudotyped lentiviral particles. ADE was seen to occur in different immune cell types, particularly in the monocytic cell lineage and the enhancement of infection in human macrophages by the presence of anti-viral antibodies. Furthermore, ADE of SARS-CoV infection was found to depend on the properties of the intracellular domain of FcγRII [109]. Consequently, the complexity of these observations should be considered, when developing new vaccines targeting SARS-CoV-2.

### 2.4. ADE in SARS-CoV-2

SARS-CoV-2 could follow an Fcγ-mediated ADE mechanism. Structural proteins S and N are likely to play a crucial role in any possible ADE, although there are not enough studies to verify this for the new virus [98]. A further study investigating antibodies against the N protein showed the opposite, suggesting the cross-reactive IgG presence is exclusive to the S protein [110]. As the SARS-CoV-2 pandemic has progressed, a compelling observation was noted in relation to the early appearance of IgG rather than IgM in certain patients [111]. This point could facilitate Fcγ-mediated ADE, and the early emergence of IgG could contribute to disease prognosis [111]. In addition, pre-existing anti-coronavirus IgG antibodies from a previous infection may cross react with SARS-CoV-2, and contribute to increased ADE and severity of COVID-19 disease [107]. There is evidence of cross-reactivity between SARS-CoV and other human coronaviruses (229E and OC43) [112]. Therefore, there might be a suggestion that any IgG induced or present from previous coronavirus infections could contribute to ADE and enhance disease progression in SARS-CoV-2. However, further understanding of the binding sites and the complexity of the immune system is crucially important as for instance, a SARS-CoV specific antibody, CR3022, was found to bind to the S protein of SARS-CoV-2, at an epitope that does not overlap with the ACE2 binding site [113]. Whilst another recent study has found a poor cross-neutralisation response between SARS-CoV and SARS-CoV-2, indicating the presence of non-neutralising antibodies that could contribute to the ADE process in SARS-CoV-2 [114]. The influence of a cross-neutralisation response is yet to be studied for SARS-CoV-2, but this phenomenon has previously been observed in other coronaviruses [98,100,115]. Consequently, further in vivo studies are required to confirm these points and to understand if antibodies could contribute to ADE following a SARS-CoV-2 infection. A further consideration for ADE in SARS-CoV-2 could be the influence of other coronaviruses [116]. The antigenic epitope heterogeneity occurring as a result of priming of infected individuals by prior coronavirus exposure could lead to the enhancement of COVID-19. This phenomenon has been observed previously, where SARS-CoV was said to be the cause of high mortality in China [117], with the priming virus a previous mild strain of coronavirus—229E [117].

Preclinical studies of inactivated vaccine candidates against SARS-CoV-2 in rodents and non-human primates demonstrated the stimulation of protective IgG responses, without sign of ADE [118]. Furthermore, sequence variability in coding genes for FcγR, along with extensive interspecies differences in the structure and function of FcγR could hinder our ability to understand data from these different animal models [119]. Passive transfer of convalescent plasma to serious cases of COVID-19 had a satisfactory safety profile, which demonstrates that IgG antibodies do not have pathogenic concerns and instead provide important clinical benefits [120]. This was substantiated by a study of 20,000 patients suffering severely from COVID-19, as they showed an adverse event incidence of 1–3% [121], while the occurrence of ADE has been linked to reduced titres of neutralising antibodies [122]. Another factor to consider when studying ADE in SARS-CoV-2 is mutations. There have been several new strains such as the Delta strain, which is around 60% more transmissible than the Alpha variant [123], and the Omicron variant, which 3x more transmissible than the Delta strain [124], as of yet though, there is as yet no clear-cut evidence of ADE in SARS-CoV-2. Furthermore, new viral strains are continuing to develop in other animal species and we need to remain vigilant for other emerging zoonotic strains and their consequences on human health. This might be due to the way the virus interacts with the immune system, the virus may be sufficiently adapted to humans [125], and if it were to interact with non-neutralising antibodies, the interaction may not be strong enough to cause ADE. Furthermore, even though ADE has previously been observed with SARS-CoV [98] and MERS-CoV [107] clinical data has not yet established a role for ADE in SARS-CoV-2 [126]. There has also been no sign of vaccine-mediated disease enhancement in the clinical trials for the COVID-19 vaccines [127,128]. The ongoing development in COVID-19 research field is additionally leading to the enhancement of our understanding of ADE. Okuya et al. [129] recently reported ADE antibodies are produced by the SARS-CoV-2 virus, after ADE antibodies were found in 41.4% of the tested COVID-19 patients. The occurrence of ADE in SARS-CoV-2 can be mediated by two different mechanisms, the FcγR and C1q, and the latter has also been previously reported in Ebola virus [84]. Due to the expression of FcγR on immune cells and the SARS-CoV-2 virus primarily targeting respiratory epithelial cells, ADE in SARS-CoV-2 is more likely to occur by the C1q mechanism found in respiratory epithelial cells [129]. Furthermore, by using convalescent-phase plasma and baby hamster kidney cells expressing FcγRs, Maemura et al. [41] found FcγRIIA and FcγRIIIA facilitated ADE infection in SARS-CoV-2. It is crucially important to extend these studies to include a larger number of COVID-19 patient samples to confirm the clinical significance of ADE in SARS-CoV-2 infection.

## 3. Conclusions and Future Considerations

This narrative review has focussed on the potential for ADE in SARS-CoV-2 in humans. This field is ever-evolving, and currently, there is limited evidence supporting ADE in COVID-19. ADE could appear following an infection with a different strain of the virus, when antibodies from a prior infection are regenerated, but cannot effectively neutralise the pathogen. FcγR-mediated entry and infection enhancement is a key mechanism that could contribute to ADE. However, the divergence between in vitro and in vivo data has added significant difficulty in our ability to understand and confirm whether a specific antibody or mechanism can contribute to the ADE process. Therefore, careful considerations should be dedicated to this phenomenon throughout the development of anti-viral antibodies. ADE mitigation strategies like targeted vaccine development, or the development of immunotherapeutics specifically targeting RBD, could be useful in the case of COVID-19 [130]. This is due to the previously aforementioned lessons learnt from SARS-CoV and MERS-CoV. Another approach could be directed to block certain epitopes on SARS-CoV-2 by glycosylation [131]. This means leaving only the RBD on the S protein free, which would only produce neutralising antibodies, preventing the development of non-neutralising antibodies and minimising the risk of ADE [131]. Alternatively, an immunofocussed approach could be taken, as previously reported for MERS-CoV [106], which works by developing therapeutics targeting only the receptor binding motifs in RBD, producing a better immune response and therefore a higher quantity of neutralising antibodies [106].

On the other hand, mAbs may be engineered to selectively target the motifs in RBD involved in the production of neutralising antibodies, minimising the chance of ADE and protecting the host from COVID-19 [132]. Additionally, the Fc region of an antibody could be engineered to reduce binding to the FcγR on cells, minimising ADE and specifically treating disease. This approach has been utilised previously in the successful generation of mAbs with preferential binding to treat another disease [133]. This methodology has also been trialled previously, where glycoengineered Fc-domain variants exhibited unique binding properties, and could effectively work against ADE [134,135]. Fab antibody fragments or single domain antibodies (nanobodies) could also be used in the development of effective treatments, as a lack of a Fc domain means there would be no risk of ADE [136]. However, this would result in these antibodies having a reduced half-life, requiring multiple doses for the same effect, or alternative engineering approaches to extend their longevity. The small size of these antibody fragments could enable them to interact with cryptic (hidden or sequestered) epitopes and inhibit virus attachment to host cells through steric hindrance [63]. These precautionary approaches, along with increasing knowledge in Fcγ pathways could provide a firm foundation for the development of effective antibody-based therapeutics without inducing ADE, as well as providing improved therapeutic efficacy.

For instance, AstraZeneca has recently announced the results of primary completion of a Phase 3 trial (PROVENT), in which 5197 people globally, with various immune deficiencies received a dose of either a mAbs-based cocktail (AZD7442) or a placebo (Clinical trial ID: NCT04625725). AZD7442 reduced risk of developing symptomatic COVID-19 by 77% and was the first antibody combination to potentially provide long-lasting protection against COVID-19 [137]. Similarly, Eli Lilly has announced the successful completion of their Phase 3 trial on 1035 participants (Clinical trial ID: NCT04427501). The study tested the efficacy and safety of LY-CoV555 (Bamlanivimab) and LY-CoV016 (Etesevimab) mAbs in preventing severe SARS-CoV-2 infection among infected high-risk ambulatory patients. The mAbs given in combination led to a lower incidence of COVID-19 related hospitalisation and death than the placebo [138]. Reaching positive outcomes from these trials will support the chance of using these mAbs to protect immunocompromised patients or elderly people who can generate insufficient natural antibodies in response to the vaccine. These antibody therapeutics have the potential to be highly beneficial for COVID-19 patients that cannot obtain full protection through currently approved vaccines. In addition, anti-SARS-CoV-2 antibodies could be highly valuable in immunocompromised patients, such as those with AIDS, or patients receiving anticancer drugs, radiation therapy, and stem cell or organ transplant.

## Figures and Tables

**Figure 1 ijms-23-06078-f001:**
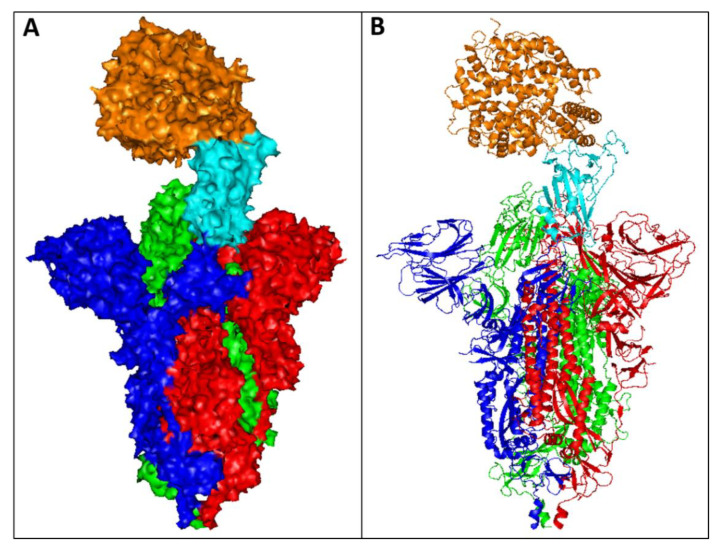
The crystal structure of SARS-CoV-2 S protein complexed with ACE2 receptor retrieved from the Protein Data Bank (PDB), PDB entry 7DF4. The structure was visualised by PyMOL (The PyMOL Molecular Graphics System, Version 1.7.4 Schrödinger, LLC.). The complex is displayed as (**A**) surface and (**B**) loops. The S protein assembles into trimers (coloured red, blue, and green) on the virion surface to form a distinctive “corona”. The RBD domain of the S protein (cyan) binds to the human ACE2 receptor (orange) to promote attachment and fusion.

**Figure 2 ijms-23-06078-f002:**
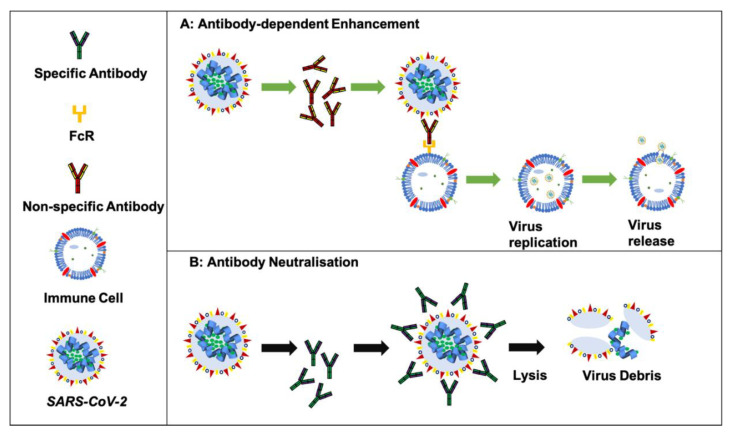
Schematic representation of antibody neutralisation versus ADE. Panel A indicates the extrinsic ADE mechanism, showing enhanced viral entry. The intrinsic ADE pathway leads to increased virus production by inhibition of the type1 interferon. Figure modified from Zhou et al. [67].

**Table 1 ijms-23-06078-t001:** List of antibodies currently in clinical development. Data were extracted from National Institutes of Health databank (https://clinicaltrials.gov/; accessed on 20 April 2022), and the Antibody Society (https://www.antibodysociety.org/covid-19-biologics-tracker/; accessed on 10 May 2022).

	Antibody	Company and Country	Clinical Trial Stage and ID
1	REGN-COV2, a cocktail of two mAbs: REGN10987 (Imdevimab) and REGN10933 (Casirivimab)	Regeneron Pharmaceuticals, Westchester County, USA	Emergency Use Authorization granted in the USA; Approved in Japan, UK, EU, and Australia
2	LY-CoV555 (Bamlanivimab) and LY-CoV016 (Etesevimab)	AbCellera, Vancouver, Canada and Eli Lilly, Indianapolis, USA	Emergency Use Authorization granted in the USA in 2020, but limited the authorisation in 2022.
3	VIR-7831/ GSK4182136 (Sotrovimab)	Vir Biotechnology, San Francisco, USA and GSK, Middlesex, UK	Emergency Use Authorization granted in the USA; Approved in Australia, UK, and EU.
4	CT-P59 (Regdanvimab)	Celltrion Group, Incheon, South Korea	Emergency Use Authorization granted in South Korea and EU.
5	AZD7442 (AZD8895/Tixagevimab and AZD1061/Cilgavimab)	AstraZeneca, Macclesfield, UK	Emergency Use Authorization granted in the USA
6	TY027	Tychan, National University of Singapore	Phase 3 (NCT04429529 and NCT04649515)
7	BRII-196/ BRII-198 (Amu-barvimab/Romlusevimab)	Brii Bio, Durham, USA/TSB Therapeutics/Tsinghua University, China	Approved in China Phase 3 (NCT04501978, NCT04479631, and NCT04479644)
8	ADG20	Adagio Therapeutics, Waltham, USA	Phase 2/3 (NCT04805671 and NCT04859517)
9	SCTA01	Sinocelltech, China	Phase 2/3 (NCT04483375 and NCT04644185)
10	C144-LS and C-135-LS	Bristol-Myers Squibb, New York City, USA	Phase 2/3 (NCT04700163 and Activ-2 study)
11	ADM03820	Ology Bioservices, Alachua, USA	Phase 2/3 (NCT05142527)
12	REGN14256 + imdevimab	Regeneron, Westchester County, USA	Phase ½/3 (NCT05081388)
13	MAD0004J08	Toscana Life Sciences Sviluppo s.r.l, Siena, Italy	Phase 2/3 (NCT04932850 and NCT04952805)
14	MW33	Mabwell Bioscience Co, Zhangjiang Hi-tech Park, Shanghai	Phase 2 (NCT04533048 and NCT04627584)
15	Etesevimab (JS016, LY3832479, LY-CoV016)	Junshi Biosciences, China and Eli Lilly, Indianapolis, USA	Phase 2 (NCT04441918, NCT04441931, and NCT04427501)
16	BGB-DXP593	BeiGene, Beijing, China	Phase 2 (NCT04551898 and NCT04532294)
17	COVI-AMG (STI-2020)	Sorrento Therapeutics, San Diego, USA	Phase 2 (NCT04734860)
18	LY-CoV1404, LY3853113	AbCellera, Vancouver, Canada and Eli Lilly, Indianapolis, USA	Phase 2 (NCT04634409)
19	IBIO-123	Immune Biosolutions, Sherbrooke, Canada	Phase 2 (Not Available)
20	VIR-7832	Vir Biotechnology, San Francisco, USA	Phase 1/2 (NCT04746183)
21	COR-101	CORAT Therapeutics, Braunschweig, Germany	Phase 1/2 trial (NCT04674566)
22	DZIF-10c, BI 767551	University of Cologne, Germany	Phase 1/2 (NCT04631666 and NCT04631705)
23	XVR011	Exevir Bio BV, Belgium	Phase 1/2 (NCT04884295)
24	HLX70	Hengenix Biotech Inc., Milpitas, USA	Phase 1 (NCT04561076)
25	DXP-604	BeiGene, Beijing, China	Phase 1 (NCT04669262)
26	ZRC-3308	Zydus Cadila, Ahmedabad, India	Phase 1 (Not Available)
27	HFB30132A	HiFiBiO Therapeutics, Cambridge, USA	Phase 1 (NCT04590430)
28	ABBV-47D11	Abbvie, North Chicago, USA	Phase 1 (NCT04644120)
29	C144-LS and C-135-LS	Bristol-Myers Squibb, New York City, USA	Phase 1 (NCT04700163)
30	JMB2002	Jemincare Group, Shanghai	Phase 1 (ChiCTR2100042150)
31	IMM-BCP-01	Immunome, Inc., Exton, USA	Phase 1 (Not Available)
32	SCTA01	Sinocelltech, China	Phase 1 (NCT04483375)
33	MW33	Mabwell Bioscience Co., Ltd., Shanghai	Phase 1 (NCT04533048)
34	Anti-SARS-CoV-2 mAb	Stanford University, Stanford, USA	Phase 1 (NCT04567810)
35	P2C-1F11	Brii Biosciences, Durham, USA	Phase 1 (NCT04479631 and NCT04479644)
36	SCTA01	Sinocelltech, China	Phase 1 (NCT04483375)
37	LY-CovMab	Luye Pharma Group, Princeton, USA	Phase 1 (NCT04973735)
38	CT-P63	Celltrion Group, Incheon, South Korea	Phase 1 (NCT05017168)
39	IGM-6268	IGM Biosciences, Mountain View, USA	Phase 1 (NCT05160402 and NCT05184218)

## Data Availability

Not applicable.

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
