# Peer review of "The Role of Antibodies in the Treatment of SARS-CoV-2 Virus Infection, and Evaluating Their Contribution to Antibody-Dependent Enhancement of Infection"

_ijms, 2022, doi:10.3390/ijms23116078_

Round 1

Reviewer 1 Report

Dear authors,

This short review is well written and discusses an elusive topic. Secondary infections caused by SARS-CoV-2 variants have been widely reported during variant waves, but ADE was never clearly demonstrated.

Minor comments are following:

As a review, I suggest the authors to refer and discuss additional articles, including among others:

Lee et al. 2020 (https://www.nature.com/articles/s41564-020-00789-5)

Yip et al 2016 (https://pubmed.ncbi.nlm.nih.gov/27390007/)

Tetro 2020 (https://www.ncbi.nlm.nih.gov/pmc/articles/PMC7102551/)

I believe few extra topics could be discussed in the manuscript:

The non-antibody-based mechanisms such as cytokine cascades and cell-mediated immune pathology observed in enhanced respiratory diseases, including cases after vaccinations with respiratory syncytial and measles viruses in the 1960's.

Evidences of ADE in animal coronaviruses. Some animals are now being vaccinated for SARS-CoV-2 as well. Some animals as mink and white-tailed deer have shown potential to spread novel animal-originated specific SARS-CoV-2 strains that could spill over to humans. The impact of a secondary infection or primary infection after vaccination caused by these and other reassortment-originated strains in humans is unknown.

As indicated in the manuscript, theoretically we could be seeing cases of ADE with the advance of vaccination. Despite some exceptions, we have an increasing vaccination coverage worldwide and no ADE cases have been reported. The treatment with mAbs is fairly discussed in the manuscript, but the potential occurrence after vaccination is less elaborated. I suggest the authors to discuss a little more the vaccination topic with extra few sentences and references.

Reviewer 2 Report

This is a very interesting review paper. The informations presented in this article are relevant to the scientific literature due to the important role played by the antibodies.

The article presents the Antibody-dependent enhancement (ADE) phenomenon, the occurrence in viral infections and also investigates its role in  the infection produced SARS-CoV-2 virus.

I would suggest the authors to specify that this is a descriptive review.

I would also recommend the authors to add more informations in the chapter 1.3. (The role of antibodies against SARS-CoV-2 virus ) regarding their role because in my opinion this is the most important chapter from all the article.

At line 225 the ahtors mention „Numerous articles have examined „ but they cite only one article.
